Methodological recommendations for assessing scleractinian and octocoral recruitment to settlement tiles

Harper Leah M. harperl@si.edu 1 2
Huebner Lindsay K. 3
O’Cain Elijah D. 4 5
Ruzicka Rob 3
Gleason Daniel F. 4
Fogarty Nicole D. 1 6
1 Halmos College of Natural Sciences and Oceanography, Nova Southeastern University , Dania Beach , FL , United States of America
2 Current affiliation:  Tennenbaum Marine Observatories Network, MarineGEO, Smithsonian Environmental Research Center , Edgewater , MD , United States of America
3 Fish and Wildlife Research Institute, Florida Fish and Wildlife Conservation Commission , Saint Petersburg , FL , United States of America
4 James H. Oliver, Jr., Institute for Coastal Plain Science, Georgia Southern University , Statesboro , GA , United States of America
5 Current affiliation:  Coastal Resources Division, Georgia Department of Natural Resources , Brunswick , GA , United States of America
6 Current affiliation:  Department of Biology and Marine Biology, Center for Marine Science, University of North Carolina Wilmington , Wilmington , NC , United States of America
Reimer James
Electronic publication date: 2021 Dec 17
Publication date: 2021
Volume: 9
Electronic Location ID: e12549
Received 2021 Jun 23; Accepted 2021 Nov 4
Copyright: ©2021 Harper et al.
Copyright year: 2021
Copyright holder: Harper et al.
License: This is an open access article distributed under the terms of the Creative Commons Attribution License, which permits unrestricted use, distribution, reproduction and adaptation in any medium and for any purpose provided that it is properly attributed. For attribution, the original author(s), title, publication source (PeerJ) and either DOI or URL of the article must be cited.
License URL: https://creativecommons.org/licenses/by/4.0/

Keywords: Recruitment, Settlement tile, Rugosity, Scleractinia, Octocorallia, Florida Reef Tract

Funding: The National Oceanic and Atmospheric Administration Coral Reef Conservation Program under National Ocean Service Agreement Codes MOA-2010-026 MOA-2015-047 This research was funded by the National Oceanic and Atmospheric Administration Coral Reef Conservation Program under National Ocean Service Agreement Codes MOA-2010-026 and MOA-2015-047. The funders had no role in study design, data collection and analysis, decision to publish, or preparation of the manuscript.

==============================
Quantifying recruitment of corals is important for evaluating their capacity to recover after disturbances through natural processes, yet measuring recruitment rates in situ is challenging due to the minute size of the study organism and the complexity of benthic communities. Settlement tiles are widely used in studies of coral recruitment because they can be viewed under a microscope to enhance accuracy, but methodological choices such as the rugosity of tiles used and when and how to scan tiles for recruits post-collection may cause inconsistencies in measured recruitment rates. We deployed 2,880 tiles with matching rugosity on top and bottom surfaces to 30 sites along the Florida Reef Tract for year-long saturations during a three year study. We scanned the top and bottom surfaces of the same tiles for scleractinian recruits before (live scans) and after treating tiles with sodium hypochlorite (corallite scans). Recruit counts were higher in corallite than live scans, indicating that scleractinian recruitment rates should not be directly compared between studies using live scans and those scanning tiles which have been processed to remove fouling material. Recruit counts also were higher on tile tops in general, but the proportion of settlement to the top and bottom surfaces varied significantly by scleractinian family. Thus, biases may be introduced in recruitment datasets by differences in tile rugosity or by only scanning a subset of tile surfaces. Finally, we quantified octocoral recruitment during live scans and found they preferentially settled to tile tops. We recommend that recruitment tile studies include corallite scans for scleractinian skeletons, deploy tiles with matching rugosity on top and bottom surfaces, and scan all tile surfaces.

Introduction

Settlement tiles are often used to evaluate variation in coral recruitment through time and space (Rogers et al., 1984; Edmunds, 2017; Davidson et al., 2019). Tiles can be removed from study sites and examined under a microscope, facilitating identification of small recruits and allowing an assessment of recruitment during a defined time period. However, various methodological options exist for conducting settlement tile studies which can influence the resultant recruitment rates, making interstudy comparisons problematic and ecological interpretations inconsistent (Table S1).

One such methodological decision that can affect quantified recruitment is the technique used to scan tiles for settled corals. Tiles can be scanned for living recruits shortly after retrieval, typical for studies estimating recruit survivorship (e.g., Arnold, Steneck & Mumby, 2010) or for sampling recruit tissue for molecular identification (O’Cain et al., 2019; Guerrini et al., 2020). Alternatively, tiles can be soaked in sodium hypochlorite to reveal corallite morphology (Harriott & Fisk, 1987), which can be used to more accurately identify recruits than is possible during live visual scans, and to remove encrusting organisms that potentially obscured recruits which experienced post-settlement mortality.

Whether to scan all tile surfaces (top, bottom, and sides) or only a subset represents another methodological choice. For most studies that have compared tile surfaces, >90% of recruits were found on the bottom surface (Green & Edmunds, 2011; Edmunds, Nozawa & Villanueva, 2014; Humanes & Bastidas, 2015). Consequently, some studies exclude the top surface from analysis (Chong-Seng, Graham & Pratchett, 2014; Burt & Bauman, 2020; Gouezo et al., 2020). However, commonly there are differences in rugosity on tile surfaces used in recruitment studies (Green & Edmunds, 2011; Edmunds, 2017; Lal et al., 2018), with tops being smooth and bottoms bearing textures which provide microhabitat refugia for recruits. Tile rugosity may mimic heterogenous natural substratum and enhance rates of scleractinian recruitment and survivorship on tiles (Nozawa, 2008; Edmunds, Nozawa & Villanueva, 2014; Randall et al., 2021); if the top surface of the tile is also textured, the proportion of settlement there can be increased (Edmunds, Nozawa & Villanueva, 2014). Thus, rugosity differences between tile top and bottom surfaces may confound recruitment data, as can scanning only a subset of tile surfaces.

Most coral reef settlement tile literature focuses on scleractinian recruitment. However, as scleractinians have declined throughout the Caribbean, octocoral densities have remained stable or increased (Ruzicka et al., 2013; Edmunds & Lasker, 2016; Lasker et al., 2020). Octocoral recruitment is typically quantified in situ (Lasker, 2013; Lasker & Porto-Hannes, 2021) and rarely on tiles, potentially leaving gaps in understanding the spatial and temporal variation in octocoral recruitment.

During a study initially designed to investigate the spatial and temporal variation in scleractinian and octocoral recruitment on 2,778 settlement tiles deployed throughout the Florida Reef Tract (Harper et al., unpublished data), we observed methodological inconsistencies in the coral recruitment literature (Table S1). We therefore designed our study to also collect companion data on our methodological choices, in an effort to critically examine the potential implications of these choices, and of those made in other studies, on the ecological results. Specifically, we present here an assessment of recruitment tile methods in which we (1) compare scleractinian recruitment rates quantified through live and corallite scans on the same tiles, (2) compare scleractinian recruitment overall and by taxonomic family to the tops and bottoms of tile surfaces with matching rugosity, and (3) quantify live octocoral recruits on the top and bottom surfaces of the same tiles. We conclude with recommendations for future coral recruitment studies on tile rugosity, and when and how to scan tiles for scleractinian and octocoral recruits.

Methods

Tiles were deployed at 30 sites along the Florida Reef Tract, from Broward County in the north to Key West in the south, with depths ranging from 1.5 to 18 m (Fig. 1, Table S2), in accordance with the following authorizations: Florida Keys National Marine Sanctuary Permit FKNMS-2013-058-A1, Department of the Army Permit SAJ-2014-01396, and Broward County Environmental Resource License DF14-1048. Tiles were unglazed terracotta (15cm×15cm×1cm ), consistent with many previous studies of coral recruitment (Mundy, 2000; Arnold & Steneck, 2011; Green & Edmunds, 2011; Edmunds, Nozawa & Villanueva, 2014; Van Woesik, Scott & Aronson, 2014; Humanes & Bastidas, 2015). Because each tile was smooth on one surface and grooved on the other, tiles were deployed in back-to-back pairs, with grooved surfaces facing outward, to provide consistent rugosity on the top and bottom of each tile pair (Figs. 2A–2B).

Figure 1 Map of Florida Reef Tract study sites.

Point color is scaled to depth in meters. Sites spanned four regions of the reef tract: Southeast Florida and the Upper, Middle, and Lower Keys, and are labeled with numbers corresponding to Table S2. Map tiles by Stamen Design, under CC BY 3.0. Data by OpenStreetMap, under ODbL.

Figure 2 Settlement tiles before, during and after deployment.

Terracotta tiles were configured in pairs, grooved sides facing outward, yielding consistent rugosity on the top and bottom surfaces of the pair. Tiles were deployed by installing a green plastic drywall anchor into a hole drilled into the substratum; stainless steel lag screws were used to fasten the tile pairs into the anchor (A). A tile pair deployed to the reef (B) and transport racks after retrieval of tiles, before surfacing (C). Tiles were submerged in trays filled with seawater for scans of living recruits (D, F), then soaked in sodium hypochlorite and scanned for corallites (E, G). Images D and E depict the same individual from the family Siderastreidae before and after tissue removal. Images F and G depict the same individual from the family Poritidae before and after tissue removal. Scale bars represent one mm. Photo credits: (A) Rob Ruzicka, (B) Lindsay Huebner, (C) Lauren Stefaniak, (D–G) Leah Harper.

At each of 30 sites (Fig. 1, Table S2), we haphazardly deployed 32 tile pairs by attaching them to the substratum. Tile pairs were arranged along four permanent transects ∼22 m in length at each site, with eight tile pairs per transect. The distance between nearest neighboring tiles was 1–10 m. Using a pneumatic drill attached to a SCUBA cylinder, we drilled holes in the substratum with a carbide drill bit (1.25–1.59 cm diameter), into which we placed 1.25 cm diameter ribbed plastic drywall anchors. Tile pairs were secured, generally parallel to the substratum, using a stainless steel lag screw (0.64 cm diameter, 5–13 cm length, depending on substratum rugosity) inserted through pre-drilled center holes in the paired tiles and screwed into the drywall anchor (Figs. 2A–2B). Tile deployments occurred from February to April in 2015, 2016, and 2017 (Table S2), allowing a minimum of four weeks for settlement tiles to condition before expected Porites astreoides planulation (Edmunds, Gates & Gleason, 2001). Tiles were retrieved approximately 12 months after deployment (range 11.5–13.5 months) in each of 2016, 2017, and 2018 (Table S2). Tiles were removed from the substratum and transported on racks made from two layers of egg crate material with bolts to hold the tiles in place and nuts to space the tiles from the racks, preventing possible damage to recruits (Fig. 2C).

The grooved top and bottom of tile pairs were submerged in trays of seawater while examined under a dissecting microscope for living scleractinian and octocoral recruits (Figs. 2D, 2F). Subsequently, tiles were submerged in 10% sodium hypochlorite solution for 24–48 h to remove live tissue, then dried for scleractinian corallite scans (octocorals do not leave skeletal material behind). During corallite scans, the grooved top and bottom surfaces were examined a second time to locate scleractinian recruit skeletons (Figs. 2E, 2G). We additionally scanned the tile sides (edges) for corallites, which was not possible during live scans with the tiles submerged in trays. To determine whether taxonomic groups of scleractinians settled in different proportions across tile surfaces, corallites were identified to the family level using diagnostic characteristics such as shape, calice diameter at budding stage, columella development, and septal arrangement and dentition (Smith, 1971; Budd et al., 2001; Budd & Stolarski, 2011; Humblet, Hongo & Sugihara, 2015). Corallites that were damaged, underdeveloped, or did not meet diagnostic criteria were classified as unidentified. Diagnostic characteristics were supported with molecular evidence using a single-step nested multiplex PCR assay designed to identify Caribbean scleractinian recruits, developed from a subset of recruits sampled during live scans (O’Cain et al., 2019). Further, we conducted blind inter-observer comparisons of corallite identifications to ensure standardization.

To determine whether scan method (live vs. corallite) influenced recruit count, we fit a negative binomial generalized linear model with a log link (GLM) in R v.4.0.3 (R Core Team, 2020) using the package MASS (Venables & Ripley, 2002). We modeled the effects of the fixed factors of site, year, location on tile, and scan method (allowing for an interaction between location and scan method) on the response variable recruit count. Model assumptions were assessed visually. We evaluated the significance of fixed effects with Type II analysis of deviance using the ‘car’ package (Fox & Weisberg, 2019). In addition, we modeled the effects of scan method, location on tile, and interaction within each year and applied Tukey’s pairwise contrasts to assess within-year differences in recruit count between groups using the ‘emmeans’ package (Lenth, 2021).

The majority (99.9%) of recruits identified to family level in the corallite scans were in the families Siderastreidae, Poritidae, Agariciidae, and Faviidae (membership in Faviidae as assigned in Budd & Stolarski, 2011). For these identified recruits, we used a negative binomial GLM with a log link to assess the effects of coral family and tile surface (top, side, and bottom) on recruit count from corallite scans. We also included year and site as fixed factors, to account for temporal and spatial autocorrelation, and included area of tile surfaces as an offset variable. Because site depth potentially drives the tile surface of scleractinian recruitment (Rogers et al., 1984), we included depth as a covariate. The model allowed for interactions between coral family, location on tile, and depth. We tested for significance of fixed effects with Type II analysis of deviance and evaluated significant differences in settlement orientation within families using Tukey’s pairwise contrast. To test the significance of individual scleractinian families’ relationships between recruitment location on tile and depth, we used simple linear regressions with site-wide proportions of each family on tops, bottoms, and sides of tiles as response variables and depth as the explanatory variable. For regressions revealing significant relationships, we calculated Pearson’s product moment correlation coefficients.

Finally, for octocorals, we used a generalized linear model (quasipoisson family with a log link) to regress total live recruit count against the factors of tile surface (top and bottom) and year, with depth as a covariate, allowing for interaction between tile surface and depth. After assessing model assumptions visually, we tested for significance of fixed effects with Type II analysis of deviance and evaluated significant differences in tile surface within each year using Tukey’s pairwise contrast. For all analyses, extreme values were trimmed from figures where necessary for readability (as reported in figure legends) but were incorporated in statistical analyses.

Results

Over three years, we deployed 2,880 tile pairs and retrieved 2,778 (96% retrieval). Most of the unretrieved tile pairs (93 of 102) were lost during the 2017–2018 deployment, when Hurricane Irma made landfall in the Florida Keys. In all three years, significantly more scleractinians were found in corallite than live scans on both the top and bottom tile surfaces (ANOVA p < 0.001; Fig. 3, Table 1), with a significant interaction between scan method and tile surface (ANOVA p = 0.008). Overall, we found 379% more recruits in corallite than live scans (excluding those found on tile sides, which were not scanned live).

Figure 3 Density of scleractinian recruits on top and bottom tile surfaces by scan method (live and corallite) across N = 30 Florida Reef Tract sites for each of three years of tile deployment.

Boxplots show mean density of recruits at each study site, and display the median (horizontal line), the first and third quartiles (interquartile range [IQR]), 1.5 × IQR (whiskers), and outliers (black dots). Two outliers with density values greater than 1,500 m−2 were trimmed from the plot for readability (2,164 m−2, 4,128 m−2; both from top surface, corallite scan, 2018). Letters denote significant pairwise difference of recruit count between top and bottom tile surfaces, as well as live and corallite scans of each tile surface, within each year (Tukey’s p < 0.05).

Corallite scans found significantly higher numbers of scleractinian recruits on top compared to bottom surfaces of tiles in each year (Tukey p < 0.001 for 2016 and 2018; Tukey p = 0.001 for 2017; Fig. 3, Table 1). When the data were pooled across all three years, our recruit community was dominated by siderastreids, which settled in vastly higher numbers on tile tops, followed by sides and bottoms (Tukey’s p < 0.001 for top-side and top-bottom contrasts; Fig. 4, Table 2). Poritids were our second most common group, and they also showed a strong preference for tile tops (Tukey’s, top-bottom p < 0.001), followed by bottoms (Tukey’s, side-bottom p < 0.001). Agariciids preferentially settled on tile bottoms (Tukey’s, bottom-side p < 0.001, bottom-top p < 0001), but were our third most common family. Faviid settlement was low overall, with significantly more recruits found on tops and bottoms than tile sides (Tukey’s, side-top p = 0.007, side-bottom p = 0.002).

Table 1 Type II analysis of deviance of a negative binomial generalized linear model regressing the number of scleractinian recruits found on the top and bottom surfaces of tiles (location) in live and corallite scans (method).

Site and year are fixed factors. The chi-square value (Chisq), degrees of freedom (df), and p-value (Pr[>Chisq]) are shown.

 	Chisq	Df	Pr(>Chisq)	
Location on tile	440.98	1	<0.001	
Method	563.02	1	<0.001	
Year	1287.02	2	<0.001	
Site	1111.11	29	<0.001	
Interaction of location & method	7.02	1	0.008	

Figure 4 Density of scleractinian recruits found on tile surfaces during corallite scans across N = 30 Florida Reef Tract sites and three years of deployments.

Boxplots show mean density of recruits at each study site; N = 90 site-years surveyed. Boxes display the median (horizontal line), the first and third quartiles (interquartile range [IQR]), 1.5 × IQR (whiskers), and outliers (black dots). Five outliers with density values greater than 100 m−2 were trimmed from the plot for readability (198 m−2, 215 m−2, 295 m−2, 442 m−2, 723m−2; all from top surface, Siderastreidae). Letters denote significant differences of recruit count within scleractinian family (Tukey’s p < 0.05).

Table 2 Type II Analysis of Deviance of a negative binomial generalized linear model regressing the number of scleractinian recruits against family, tile surface, and depth.

Year is a fixed factor. The chi-square value (Chisq), degrees of freedom (df), and p-value (Pr[>Chisq]) are shown.

Factor	Chisq	Df	Pr(>Chisq)	
Family	993.87	3	<0.001	
Tile surface	1001.04	2	<0.001	
Depth	2.98	1	0.084	
Year	478.82	2	<0.001	
Family × Tile surface	1044.3	6	<0.001	
Family × Depth	56.76	3	<0.001	
Tile surface × Depth	28.57	2	<0.001	
Family × Tile surface × Depth	26.16	6	<0.001	

Tile surface interacted with site depth and coral family (ANOVA p < 0.001, Table 2), though depth alone did not have a significant effect on the number of recruits. Simple linear regression determined that the proportion of siderastreid recruits on the tops of tiles increased with increasing depth (p = 0.042, r = 0.24; Fig. 5). Additionally, the proportion of faviids (p = 0.028, r = 0.33) and poritids (p < 0.001, r = 0.37) that settled on the sides of tiles increased with increasing site depth (Fig. 5).

Figure 5 Relationships between site depth in meters (x-axes) and proportion of corallites found on each tile surface (y-axes) per scleractinian coral family.

Significant linear relationships are denoted with asterisks (* denotes p < 0.01; *** denotes p < 0.0001) and the Pearson product-moment correlation coefficient is shown for significant relationships.

Live octocoral recruit counts varied significantly by year, and tile surface (ANOVA, year p < 0.001, tile surface p = 0.003; Fig. 6, Table 3), with site-level density ranging from 0 m−2 to 182.6 m−2. In 2016, mean site-level octocoral recruit density was 3.8 ± 1.8 m−2, but mean densities rose to 16.6 ± 6.3 m−2 in 2017 and 8.1 ± 4.0 m−2 in 2018. Overall, 67% of octocorals were on tile tops, though the pairwise within-year difference between top and bottom surfaces was only significant in 2018 (Tukey’s, 2018 top-bottom contrast p < 0.001). While overall octocoral recruitment did not vary with site depth, the interaction between depth and location on tile was significant (p < 0.001); Table 3.

Discussion

In our three year study of coral recruitment to the Florida Reef Tract, we counted scleractinian recruits in both live and corallite scans and found significantly more recruits in corallite scans overall. While others have acknowledged that removing tissue of corals and fouling organisms may result in a higher recruit count (Arnold & Steneck, 2011), this difference has rarely been quantified. We attribute the difference between corallite and live scan counts to recruits that were overgrown, smothered by sediment, otherwise dead, and/or very small (<0.5 mm) and likely obscured during live scans. The significant interaction between scan method and tile surface may reflect the difficulty in locating small recruits during live scans on heavily sedimented top surfaces of tiles. This large discrepancy in numbers on the same tiles indicates that scleractinian recruitment rates should not be directly compared between studies only performing live scans and those quantifying recruitment on tiles that have been post-processed to remove fouling material. These results also suggest that for studies focusing on overall scleractinian recruitment without concern for the effect of post-settlement survivorship, soaking tiles in sodium hypochlorite provides a more accurate estimate than live scans.

Figure 6 Density of octocoral recruits found on top and bottom tile surfaces during live scans across N = 30 Florida Reef Tract sites for each of three years of deployment.

Boxplots show the mean density of recruits at each study site within each year. Boxes display the median (horizontal line), the first and third quartiles (interquartile range [IQR]), 1.5 × IQR (whiskers), and outliers (black dots). Two outliers with density values greater than 200 m−2 were trimmed from the plot for readability (240 m−2; top surface 2018, and 340 m−2; top surface 2017). Asterisks denote significant pairwise differences of recruit count within year (*** denotes p < 0.001).

Table 3 Type II analysis of deviance of a quasipoisson generalized linear model regressing the number of octocoral recruits against tile surface and depth.

Year is a fixed covariate. The chi-square value (Chisq), degrees of freedom (df), and p-value (Pr[>Chisq]) are shown.

 	Chisq	Df	Pr(>Chisq)	
Location on Tile	43.56	1	<0.001	
Depth	3.06	1	0.081	
Interaction of Location & Depth	82.35	1	<0.001	
Year	120.29	2	<0.001	

Contrary to most assessments of scleractinian settlement orientation in shallow water, we found the majority of recruit corallites on the top surfaces of our settlement tiles in every year of our study (Fig. 3). Scleractinians have been reported to settle predominantly on bottom surfaces of tiles because that side of the tile mimics the natural cryptic microhabitats preferred for settlement (Green & Edmunds, 2011; Humanes & Bastidas, 2015). Further, studies that include sites deeper than 10m tend to reveal a shift in settlement orientation from tile bottoms and sides to tops with increasing depth (Rogers et al., 1984; Babcock & Mundy, 1996). While our study identified some shifts in the proportion of recruits that settled to tile top (siderastreids) and side surfaces (poritids and faviids) with increasing site depth, these shifts were modest and not consistent across the coral families (Fig. 5). Because increasing site depth does not explain the overall majority of settlement to tile tops observed in our study, we propose that including rugosity on tile tops in this study may have enhanced recruitment to that surface (Nozawa, Tanaka & Reimer, 2011; Edmunds, Nozawa & Villanueva, 2014). Thus, biases are introduced in recruitment studies when top and bottom tile surfaces differ in rugosity and/or quantification is limited to tile bottoms.

Regardless of site depth, the proportion of settlement to top and bottom surfaces varied by scleractinian family. Siderastreids and poritids settled predominantly on tile tops, while agariciids settled predominantly on tile bottoms. Therefore, biases introduced by differences in tile rugosity or by only scanning tile bottoms may over- or underestimate the relative contributions of scleractinian families to overall recruitment. Agariciids usually settle cryptically (Morse et al., 1988) and often dominate Caribbean recruit communities (Rogers et al., 1984; Arnold & Steneck, 2011). Had we only quantified recruits on tile bottoms, we would have concluded that agariciids were the dominant member of our recruit community, consistent with prior studies. We also would have significantly underestimated recruitment by siderastreids, reducing it to a level comparable to that of faviids. Thus, our decision to scan tile tops in addition to bottoms has profound implications for any ecological conclusions on the relative recruitment of these coral families in our study region.

As with scleractinians, we found more octocoral recruits on tile tops (67% of the three year total) compared to bottoms; although this difference was only significant in 2018, it emphasizes the importance of scanning all tile surfaces. This majority of recruits on the top surfaces is unlikely to be explained by a survivorship advantage there, as studies which have deployed lab-reared recruits in situ have found low survivorship on tile tops (Lasker, Kim & Coffroth, 1998; Evans, Coffroth & Lasker, 2013). Without the ability to measure overall recruitment (i.e., conducting a corallite scan) as can be done for scleractinians, counts of living octocoral recruits on tiles, regardless of surface, include some level of post-settlement mortality. The abundance per survey area of octocoral recruits in situ is highly variable, making the assessment of post-settlement mortality on octocoral recruit densities difficult (Lasker & Porto-Hannes, 2021), compounded with the difficulty of seeing very small recruits. However, tiles can be examined under a microscope, likely allowing for smaller recruits to be counted than through in situ surveys. The densities of live octocoral recruits in our study were on par or higher than those that have been found during in situ surveys (Privitera-Johnson, Lenz & Edmunds, 2015; Lasker & Porto-Hannes, 2021), indicating the viability of settlement tiles for the study of octocoral recruitment. While assessments of recruit survivorship were beyond the scope of our study, researchers may be able to use tiles to more clearly assess the effects of post-settlement mortality on octocoral recruitment than can be done during in situ surveys. For example, tiles can be collected, scanned for even very small living recruits under a microscope, and redeployed for repeated live recruit scans, as has been done for scleractinians (Arnold, Steneck & Mumby, 2010; Arnold & Steneck, 2011)

Conclusions

Our three-year, region-wide recruitment study provided an opportunity to evaluate methodological considerations that may confound efforts to compare coral recruitment rates between studies that used different methods. First, we found higher densities of scleractinian recruits in corallite than live scans, indicating that while live scans can have useful applications such as quantifying post-settlement survival or allowing for molecular identification of recruits, they are likely to underestimate overall scleractinian recruitment rates when directly compared to studies that examine tiles treated with sodium hypochlorite. We recommend that, for comparisons of global scleractinian recruitment, tiles should be scanned after being processed for the removal of sediment and benthic organisms that could obscure the detection of small recruits. Second, we found more scleractinian recruits on tile tops than bottoms, indicating that rugosity should be uniform on both surfaces, to ensure that differences in recruitment rates to these surfaces are due to ecological and not methodological factors. Further, scanning all surfaces of the tiles will avoid mischaracterizing the relative contribution of identified scleractinian recruit taxa to the overall recruitment rate. Finally, we propose that tiles also can be used to enumerate living octocoral recruits, and studies that do so should abide by the same recommendations for uniform tile rugosity and scanning both tops and bottoms of the tiles.

Supplemental Information

Supplemental Information 1 Selected recruitment studies from 2007–2020 (ordered chronologically)

Recruit density reflects total scleractinian recruitment unless otherwise specified, and densities and percentages are reported as a range by site-deployment wherever possible. These examples illustrate difference methodological choices made during scleractinian recruitment studies and the discrepancies in reporting convention that may render interstudy comparisons problematic.

Click here for additional data file.

Supplemental Information 2 Characteristics of study sites (listed north to south). Site numbers correspond with those displayed in Fig. 1

Dates (dd/mm/yyyy) for each year (Yr 1–3) of tile deployment and retrieval are shown.

Click here for additional data file.

Supplemental Information 3 Raw coral recruitment data from corallite scans (after soaking tiles in sodium hypochlorite)

Recruit counts are inclusive of corals also identified in live scans. Scleractinians are identified to family, octocorals are not identified to higher resolution. Sites are classified by depth (categorical; shallow <10 m, deep >10 m), habitat, and region (SEFL = southeast Florida). “Loc” indicates location on tile (top, bottom, or side). Taxon columns indicate families as follows: “AGAR” = Agariciidae, “FAVI” = Faviidae, “PORI” = Poritidae, “SIDE” = Siderastreidae, “Other” = identified corals outside of the four previous families, “UNKS” = unidentified scleractinian recruits, “TotalSto” = total scleractinian count, ”Total Oct” = total octocoral count, “TotalCor” = total coral count.

Click here for additional data file.

Supplemental Information 4 Raw results of live scans of tiles before soaking in sodium hypochlorite

Recruit counts of corals found in live scans. Data are classified by year, site, tile, and top or bottom of tile. Coral recruits are identified as stony coral or octocoral under the column “variable”. “Value” indicates the raw count of scleractinians and octocorals. “LiveArea” indicates density of live corals on each tile surface (transformed by dividing the count by the area).

Click here for additional data file.

Supplemental Information 5 Site metadata table used to produce map figure and depth analyses

This is the final site metadata file produced within the submitted code. Site number is generated by merging with the numbers list (also submitted), and depth was transformed from feet to meters. The column “Reef.Type” indicates whether the site is located on nearshore hardbottom or inner, middle, or outer terraces (southeast Florida), or whether the site is deep or shallow forereef or an inshore patch reef (FL Keys). The latitude and longitude columns have been manually nudged to allow sites that are very close together to be visibly mapped. Columns “True.Lat” and “True.Long” were added before submission; these represent the unedited latitudes and longitudes of study sites.

Click here for additional data file.

Supplemental Information 6 Site numbers ordered from by latitude from north to south, used within our script to assign numbers to map figure (Fig. 1) to be compatible with Table 2

“No” = site number in order of decreasing latitude.

Click here for additional data file.

Supplemental Information 7 R code used to produce figures and analyses reported in this manuscript

Click here for additional data file.

We thank the many Florida Fish and Wildlife Research Institute staff and Nova Southeastern University graduate students who deployed, retrieved, and live-scanned settlement tiles.

Additional Information and Declarations

Competing Interests

Author Contributions

Field Study Permissions

Data Availability

The authors declare there are no competing interests.

Leah M. Harper performed the experiments, analyzed the data, prepared figures and/or tables, authored or reviewed drafts of the paper, and approved the final draft.

Lindsay K. Huebner and Elijah D. O’Cain performed the experiments, authored or reviewed drafts of the paper, and approved the final draft.

Rob Ruzicka, Daniel F. Gleason and Nicole D. Fogarty conceived and designed the experiments, performed the experiments, authored or reviewed drafts of the paper, and approved the final draft.

The following information was supplied relating to field study approvals (i.e., approving body and any reference numbers):

This research was authorized by Florida Keys National Marine Sanctuary, Department of the Army, and Broward County.

The following information was supplied regarding data availability:

The raw data are available in the Supplemental Files.

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
