# Peer review of "Methodological recommendations for assessing scleractinian and octocoral recruitment to settlement tiles"

_PeerJ, doi:10.7717/peerj.12549_

## Round 0.1 · original submission · Major Revisions

I have heard back from two expert reviewers. One is uncertain of the interpretation of your results, while the other feels the organization of the paper, with your design and work not matched by the subsequent analyses and conclusions. In short, these constructive comments will require a major revision to address.

Reviewer 1 ·

Basic reporting

Please see my comments in the comments for authors.

Experimental design

Please see my comments in the comments for authors.

Validity of the findings

Please see my comments in the comments for authors.

Additional comments

This study examined the annual recruits of both hard and soft corals for three years at 30 sites along the Florida Reef Tract. While the study design is good and the scale of this study using a total of 2880 tiles at 30 sites over three years is large, the purpose of this study is not well-stated at the introduction and hence not clear. Especially I confused when I read the conclusion of this manuscript that focuses on the methodological aspect of this study, recommending to use corallite count, because the study design appears to be created for an ecological survey on the recruitment pattern of hard and soft corals, not for the assessment of counting methods. As the main finding (?) of higher number of corallites than live recruits is not surprising, I think it’s more meaningful to focus on the recruitment pattern of hard and soft corals found in this study. Because of this confusion I can see in the current manuscript, my recommendation on the manuscript is “major revision”. Please see below comments for improving your manuscript.

<comments>

Introduction
In the final paragraph of introduction, please state the purpose of this study. You can state them clearly here, conduct appropriate analyses to examine the purposes, and discuss them in the discussion. Any analyses and discussions that do not match with the purpose should be removed from the manuscript as they create confusion.

Materials and methods
In the paragraph of GLM; there are several GLMs employed in the analysis. However, the current descriptions on each GLM is not clear enough and confusing. Please clearly describe each GLM in this paragraph; What variables were used as a response variable, fixed factors, and covariate factors; What statistical distribution was used for each response variable with what ling function?
In the model, you should nest the groups of tiles deployed along the same permanent transect. So there are four tile groups at each site.
Year (2016, 2017, 2018) should be included as a fixed factor in the GLMs for hard coral recruits. I noticed that you did this for soft coral recruits and you should do the same for hard coral recruits.

Results
The statistical test used here should not be called “ANOVA (analysis of variance)”. It should be called “analysis of deviance” when you examined statistical significance using GLMs.
In the Fig. 1, and Table 3 and 4, water depth of site was categorized to 5, 10, 15m in Fig. 1 and >10m vs <10m in the tables. As you have the exact depth data and you use GLMs, it’s no point to create this rough categorization. In Fig.1, you can use a color gradation to show the exact depth. In the analyses on the depth effect, you can use GLMs to examine using the original depth data, not using the rough two depth categories.
I’d like to suggest to include some more analyses as you have a good dataset that allow you to examine. (1) where did the coral recruits settle on the plate surface (micro-habitat); on the exposed plate area or inside the groove?
(2) The data of live recruits vs dead recruits (only corallites) can be used to estimate post-settlement survival pattern. You can examine the effect of depth, orientation of plate surface (top vs bottom), micro-habitat on the plate surface (exposed area vs inside grooves) and the interaction of depth x surface orientation on the proportion of live vs dead.

Discussion
The 2nd paragraph; Here you discuss the unique pattern of coral recruits on top vs bottom tile surfaces along depth gradient in this study. I think it’s interesting to examine the effect of coral taxa (family) on this pattern, as this pattern might be created by different settlement patterns of recruits among the dominant coral taxa (family). Is there the depth effect on the number of recruits in each coral family? Is there any difference in the pattern of shift from top to bottom or bottom to top among the coral taxa? To analyze this, you can create a GLM for the proportion of recruits on the top surface with “coral taxa”, “depth”, and the interaction term Coral taxa x Depth as fixed factors.
Line 215; you said that “we found most (67%) octocoral recruits on tile top”, without good statistical evidence. It’s only in 2018 and no difference in the other two years. Rewrite the discussion of this paragraph.
Line 239-242; I think the purpose of live scan and corallite scan is different. So if you include advantage and disadvantage of each census method in recommendation, it would be better.

Figure and tables
Fig. 1; can you match the information between Fig. 1 and Table 1? For example, you can use the same site id number in both Fig. 1 and Table 1.
Table 2; Year needs to be considered as a fixed factor.
Table 3 and 4; It’s better to use the exact water depth using GLMs than the two depth categories. You can present the result as a regression figure with water depth for the x-axis and response variable for the y-axis.
Table 6; Why water depth was not included in this analysis? The same full analysis for hard coral recruits should be done for octocoral recruits.

Supplementary data
Th site lists; depth needs to use the unit, meters. What does the site number in the first column mean?
The format of dates in the time line file in supplementary varies and need to unify the style.
Those supplementary files need to be self-explanatory, but many labels of variables might need to have explanations. As you provide Table 1, and overlapping information in some supplementary files needs to be removed.

·

Basic reporting

The manuscript is well written and follow a standard organisation. The introduction highlights well the need of the study, the figures are well designed and clear (but see comment for the captions), and references are wisely used when needed. Regarding data accessibility, all data and code is provided in the supplementary.

Experimental design

The authors deployed and analyzed an impressive number of recruiting tiles across a large number of sites. The authors compared the count of recruits with and without treatment. They also compared the count on the top and bottom surface of the tiles for scleractinians and octocorals. The study was repeated across three years.
Congrats to the authors for the hard work.
Such large deployment of recruiting tiles is rare and the lack of comparision between the different methods of assessing recruits (from the type of tiles deployed, ie different rugosity, to the method used to count recruits) makes difficult or even prone to misinterpretations previous studies. This study aimed at filling this gap and provide important information for the information of past findings and for future work.

Validity of the findings

The experiment is well replicated, the statistics are adequate and the results (mostly) support the conclusions.

My only comments is the interpretations of the results regarding the prevalence of recruits on the top rather than on the bottom surface of the tile, as this conclusion seems to be supported mostly by the results obtained in 2018 and not so much by the results obtained in 2016 and 2017. I would appreciate if the authors could provide a more detailed explanation of their generalization as this is a main discussion points. It also suggests that previous studies may have underestimated coral recruitment. Following this result, the author then argue that previous studies could have found larger recruit counts on the bottom surface because of its higher rugosity and that here they found a different result as they controlled for rugosity. This result is therefore important and the difference found across years should, in my opinion, be discussed.

Additional comments

Methods:

line 94 : distance between nearest tiles was 1~10m. Please provide in addition the length of the different transects and perhaps the mean distance among tiles.

Results:
line 156: "The interactions between scan method and tile surface was significant". It is difficult to understand the interaction from Figure 3 and this result is not discussed. Perhaps it would be better to use Tukey groups to denotes the difference across top and bottom count too. Generally, it seems that the difference between corallite scan and live scan was greater on the top of the tile (correct me if I am wrong). Do you think it is because the surface of the tile is more prone to biofouling and sedimentation that would have hide the recruits?

Discussion
line 193: The author states and this is an important point of the paper, that most scleractinians recruits were found on the top surface of the tiles. But this seems to be especially true in 2018, and not really in the other years. Similarly for the octocorals, the difference is not really pronounced except for 2018 (the only one that is significant).
The number of scleractinians recruits is also the highest in 2018 and this could explain the large percentage of recruits found on the top of the surface. The authors should therefore discussed why 2018 show this high recruitment and why they generalize that recruits are found in higher number on the tile surface (contrary to other studies) regardless of the results obtained in 2016 and 2017.
line 210: it sounds as if you suggest that Rogers et al. overestimated the Agariciids proportion of coral recruits? Is that correct? Are there other studies that either support Rogers et al. conclusions or yours? (sorry I don't have access to that paper now)

Figures:

Figure 3: The caption is confusing in my opinion as I was not sure to which data point you refer to. Perhaps better to simply describe the figure as "boxplots of the mean densities of recruits at each study site".

Figure 4: same comments as above

---

## Round 0.2 · Minor Revisions

I have heard back from one reviewer, and they, like me, find your new paper to be much better. There are only some minor comments left to address, and I look forward to seeing your revised version soon.

Reviewer 1 ·

Basic reporting

The revised manuscript is well-written and I do not have much comment to add. I now agree that the revised manuscript is ready for acceptance.

Here are some minor problems and comments authors need to fix and consider.

Fig. 1; A scale bar is useful to understand spatial scale.

Fig. 3; Year should be displayed on bottom of each diagram.

Table 1 and 2 can be moved to supplementary.

I cannot see a good reason why R2 values need to be included in those analyses of deviance test tables. My understanding of R2 value is to show a degree of correlation between two factors of interest or to show a degree of estimation for a multivariable regression based on the best combination of factors of interest. In this study’s case, R2 seems to be calculated for the latter purpose, but why we need to know R2 value to show what?

Experimental design

no comment

Validity of the findings

no comment

Additional comments

no comment

---

## Round 0.3 · accepted · Accept

I am pleased to accept this work for publication - and thank the authors for their perseverance in seeing the revisions through. I look forward to seeing the published version.